# Low-Carbohydrate Diet among Children with Type 1 Diabetes: A Multi-Center Study

**DOI:** 10.3390/nu13113903

**Published:** 2021-10-30

**Authors:** Vit Neuman, Lukas Plachy, Stepanka Pruhova, Stanislava Kolouskova, Lenka Petruzelkova, Barbora Obermannova, Jana Vyzralkova, Petra Konecna, Jan Vosahlo, Martina Romanova, Marketa Pavlikova, Zdenek Sumnik

**Affiliations:** 1Department of Pediatrics, 2nd Faculty of Medicine, Charles University and Motol University Hospital, CZ-15006 Prague, Czech Republic; Lukas.Plachy@fnmotol.cz (L.P.); SPruhova@gmail.com (S.P.); stanislava.kolouskova@lfmotol.cuni.cz (S.K.); lenkapetruzelkova@gmail.com (L.P.); obermannova@seznam.cz (B.O.); zdenek.sumnik@lfmotol.cuni.cz (Z.S.); 2Department of Pediatrics, University Hospital Brno, CZ-62500 Brno, Czech Republic; vyzralkova.jana@fnbrno.cz (J.V.); konecna.petra3@fnbrno.cz (P.K.); 3Department of Pediatrics, 3rd Faculty of Medicine, Charles University and Kralovske Vinohrady University Hospital, CZ-10034 Prague, Czech Republic; jan.vosahlo@email.cz (J.V.); martina.romanova@fnkv.cz (M.R.); 4Department of Probability and Mathematical Statistics, Faculty of Mathematics and Physics, Charles University, CZ-18675 Prague, Czech Republic; marketa@ucw.cz

**Keywords:** low-carbohydrate diet, time in range, type 1 diabetes

## Abstract

Aims/hypothesis: The proportion of children with type 1 diabetes (T1D) who have experience with low-carbohydrate diet (LCD) is unknown. Our goal was to map the frequency of LCD among children with T1D and to describe their clinical and laboratory data. Methods: Caregivers of 1040 children with T1D from three centers were addressed with a structured questionnaire regarding the children’s carbohydrate intake and experience with LCD (daily energy intake from carbohydrates below 26% of age-recommended values). The subjects currently on LCD were compared to a group of non-LCD respondents matched to age, T1D duration, sex, type and center of treatment. Results: A total of 624/1040 (60%) of the subjects completed the survey. A total of 242/624 (39%) subjects reported experience with voluntary carbohydrate restriction with 36/624 (5.8%) subjects currently following the LCD. The LCD group had similar HbA1c (45 vs. 49.5, *p* = 0.11), lower average glycemia (7.0 vs. 7.9, *p* = 0.02), higher time in range (74 vs. 67%, *p* = 0.02), lower time in hyperglycemia >10 mmol/L (17 vs. 20%, *p* = 0.04), tendency to more time in hypoglycemia <3.9 mmol/L(8 vs. 5%, *p* = 0.05) and lower systolic blood pressure percentile (43 vs. 74, *p* = 0.03). The groups did not differ in their lipid profile nor in current body height, weight or BMI. The LCD was mostly initiated by the parents or the subjects themselves and only 39% of the families consulted their decision with the diabetologist. Conclusions/interpretation: Low carbohydrate diet is not scarce in children with T1D and is associated with modestly better disease control. At the same time, caution should be applied as it showed a tendency toward more frequent hypoglycemia.

## 1. Introduction

Type 1 diabetes (T1D) requires frequent insulin administration, intensive self-monitoring of blood glucose and daily control of nutrients intake to achieve tight metabolic control. Yet, the globally recommended targets are often not met, especially in children and adolescents [1]. The call is therefore strengthening for some sort of adjunctive therapy that would allow more patients achieve the metabolic goals. Most effective strategies include technological solutions [2,3], pharmacotherapy [4] or dietary interventions, including low carbohydrate diet (LCD) [5,6].

According to the latest International Society for Pediatric and Adolescent Diabetes (ISPAD) guidelines, the dietary recommendations for children with T1D are similar to those for general population with 45–50% of daily energy intake from carbohydrates [7]. Despite this, an alternative approach is emerging, suggesting carbohydrate restriction in the form of low carbohydrate diets (LCD). The definitions of the LCD differ and are especially challenging for the childhood. A review by Seckold et al. suggested a classification based on mean estimated energy requirements for the age of the child [8]. According to the ADA, 130g carbohydrates per day is an average minimum requirement for adults [9]. A special subgroup of very-low carbohydrate diet (VLCD) is also often defined [8,10].

The supporters of these low-carbohydrate diets argue that lower dietary carbohydrate intake eliminates postprandial hyperglycemia, decreases glycemic variability and through lower insulin dose decreases the risk of hypoglycemia [11]. In adults, several studies provide indirect evidence of the desirable effects of the LCD. Krebs et al. [12] performed a small randomized trial with long established T1D patients in which the LCD led to an improvement of their HbA1c but did not significantly affect their body weight nor continuous glucose monitoring (CGM) outcomes. In a Danish study with crossover design 10 T1D adults on sensor augmented pump therapy spent more time in range, less time in hypoglycemia and had lower glycemic variability during a week on LCD [13]. Less time in hypoglycemia and lower glycemic variability was observed on LCD by the same group of authors in a study with 14 T1D adults and 12-week duration of the study periods [14].

The evidence in children with T1D is much scarcer. In a much-cited survey by Lennerz et al. [15], 131 parents of children with T1D on LCD self-reported excellent metabolic control, high satisfaction with diabetes management and remarkably low rate of acute complications. The design of the study nevertheless does not allow us to draw any strong conclusions as mentioned in a comment by Mayer-Davis et al. [16]. On the other hand, serious concerns are often raised about the lipid profile and cardiovascular risk of patients on the LCD. Increased cardiovascular risk lipid profile and hindered growth were observed in children following the LCD [17]. Excessive weight loss and growth retardation were described in patients with epilepsy treated with ketogenic (i.e., very strict low carbohydrate high fat) diet [18,19]. On top of that, it might be that the patients on the LCD are at increased risk of severe hypoglycemia as their response to glucagon is blunted [20]. It is also necessary to mention that dietary restrictions, often imposed by the parents, might lead to diabetes distress and diabetes related family conflict which worsen the quality of life [21,22]. Dietary restrictions also increase the risk of the eating behavior disorders development [23] which were shown to increase morbidity and mortality of T1D patients [24].

As of today, there are only limited data on the prevalence of LCD use among children with T1D, let alone on its safety and efficacy. The aim of this study is to map the frequency of the LCD use among pediatric patients with T1D and retrospectively compare their metabolic and clinical features to their matched T1D controls without such dietary restrictions.

## 2. Research Design and Methods

### 2.1. Study Population Characteristic

The setting of this study were the three largest tertiary referral centers for pediatric diabetes in the Czech Republic: two in Prague (University Hospital Motol and University Hospital Kralovske Vinohrady and one in Brno (University Hospital Brno). As of May 2020, at the course of the study, these centers collectively provided care for 1258 children and adolescents with T1D.

The desired contact information (e-mail) was available for a total of 1040/1258 (83%) parents/caregivers of patients with T1D diagnosed according to ADA criteria [25] in the study centers. The study flowchart is detailed in Figure 1. All available parents/caregivers were offered participation in a form of a structured electronic survey regarding their childrens’ dietary habits. All of the respondents previously signed a written consent with the data collection and analysis for the ČENDA Registry, including anonymized secondary research. An individual electronic consent was obtained from the survey respondents for data analysis and identified matching to the registry data.

### 2.2. Assessment of Dietary Habits

Parents/caregivers were asked to report a three-day record on carbohydrate intake of their child/children with T1D and report their experience with LCD. Subjects were asked whether they voluntarily and significantly reduced their dietary carbohydrate intake (i.e., followed LCD) on a regular basis either in the present or the past (at least 3 months). LCD subjects were considered those who claimed following the LCD and their referred three-day average carbohydrate intake was <26% of the daily age-specific recommended energy intake from carbohydrates [8]. As this criterion approximately equals 130 g per day at 11 years, for children above 11 years 130 g of carbohydrates per day was considered an upper limit of LCD [26]. The respondents who admitted voluntary carbohydrate restriction but did not fulfill the criteria for total LCD were considered keeping a partial LCD and excluded from further analysis. We further defined a sub-group of subjects following the very low carbohydrate diet (VLCD), i.e., below 50 g per day or <10% of daily energy intake from carbohydrate for children below 11 years [8,10]. The second part of the questionnaire consisted of nine multiple choice questions in which the respondents provided information on the initiation of LCD, sources of information and subjective changes observed after the start of LCD. Another two questions concerning the reasons for LCD termination were asked of the patients who were following the LCD in the past.

### 2.3. Clinical and Laboratory Data

The clinical and laboratory data were obtained from children currently on the LCD and their matched controls (see below). Clinical data on the subjects age, sex, age at the T1D onset, type of therapy (multiple daily insulin injection, MDI or continual subcutaneous insulin infusion, CSII) and the center of therapy as well as HbA_1_c (last available and yearly average), three days average of bolus and basal insulin doses, body height and weight and blood pressure were obtained from the ČENDA Registry [3,27] for the last visit preceding the collection of the survey. Standard deviation scores for body height, weight and BMI were calculated from the population-based data [28] as well as the of the blood pressure percentile [29]. Lipid profile (total cholesterol, triglycerides, LDL cholesterol and HDL cholesterol) and data from continuous glucose monitors (CGM) were obtained from the subjects’ medical records assessed by the investigators. CGM data taken from the last 14 days before the last visit before the survey distribution were downloaded using a specialized software (Diasend, LibreView, CareLink) and the values for standard CGM metrics [30] were used for the analysis—time in the target range (TIR) (between 3.9–10.0 mmol/L), time in level 1 (between 3.0–3.8 mmol/L) and level 2 (below 3.0 mmol/L) hypoglycemia, time in level 1 (between 10.1–13.9 mmol/L) and level 2 (above 13.9 mmol/L) hyperglycemia, average glycemia (AG), coefficient of variation (CV) and standard deviation of glycemia (SD).

### 2.4. Statistical Analysis

Continuous data are presented as medians with interquartile range (IQR). Categorical data are summarized using absolute and relative frequencies. Comparisons of the responder set to the non-responders and of the VLCD group to the LC group were carried out using Wilcoxon two-sample test for continuous variables and Fisher’s exact test for categorical variables.

For more detailed comparison, the subjects currently on LCD were randomly matched with respondents who had no experience with LCD. The matching variables were center, sex, treatment type (MDI or CSII), age at the time of survey collection and age of T1D onset. For age the allowed difference was 1 year for patients younger than 12 years, 2 years for those aged 12–14.99 and 2.5 years for those aged 15 and more. For the age at T1D onset the difference was a maximum of 2 years for patients younger than 10 years and 4 years for 10 years and older. Patients’ characteristics and measurements were then compared using paired Wilcoxon signed rank test for continuous variables and McNemar test for categorical variables.

## 3. Results

In total, 624/1040 (60%) fully completed the questionnaire (Figure 1). Demographic and clinical characteristics of all responders compared to non-respondents are shown in Appendix A; altogether the respondents were younger (12.2 vs. 13.3 years, *p* = 0.002), had shorter duration of T1D (4.5 vs. 5.2 years, *p* = 0.008), were more often on CSII (49.6 vs. 34.5%, *p* = 0.03) and had lower HbA1c (50.0 vs. 54.5 mmol/mol, *p* < 0.001) than the non-responders. The sexes were distributed equally (*p* = 0.69).

### 3.1. Frequency of the LCD among the Responders

A total of 242/624 (38.7%) of the subjects claimed they had experience with voluntary reduction of their dietary carbohydrate intake. These include the subjects who reported experience with partial LCD (i.e., low-carbohydrate breakfast) in the present or the past (129/624, 20.7%) and those who claimed following total LCD either in the present or the past (113/624, 18.1%). At the time of the survey 36/624 (5.8%) subjects were on total LCD. Further 31/624 (5.0%) subjects reported following total LCD in the past (with referred carbohydrate intake fulfilling the criteria) but decided to end it before the time of the survey. The remaining 46/624 (7.4%) subjects claimed following total LCD but their referred carbohydrate intake did not fulfill the ADA criteria [26] for LCD. Five of the 36 subjects on the LCD (13.8%) followed the VLCD.

The frequency of LCD was not distributed equally among the centers with centers from Prague having 6.0% and 6.1% frequency of LCD subjects while the center in Brno showed only 2.4% frequency (Fisher Exact test *p* = 0.04). Total LCD was more frequent in females with 25 subjects vs. 11 male subjects (Fisher Exact test *p* = 0.02).

### 3.2. Comparison of LCD Subjects to Their Non-LCD Matched Controls

The results are shown in Table 1. The subjects on LCD started with the diet at the median of 11.2 years (95% CI 8.2–12.4 years) and kept the LCD for a median of 1.1 years (95% CI 0.6–1.9 years). The subjects on LCD had lower doses of bolus insulin (10.0 vs. 21.5 U/day, *p* < 0.001) but similar basal doses (12.0 vs. 13.5 U/day, *p* = 0.76). The total insulin daily dose per kilogram body weight was consequently lower in the LCD group (0.6 vs. 0.8 U/kg/day, *p* < 0.001).

The LCD subjects did not differ from their matched non-LCD peers in their last measured HbA_1_c (45.0 vs. 49.5 mmol/mol; 6.3 vs. 6.7%, *p* = 0.11), although the yearly average of their HbA_1_c was borderline lower in the LCD group (47.9 vs. 50.9 mmol/mol; 6.5 vs. 6.8%, *p* = 0.05). The LCD group showed higher time in target range (74.0 vs. 66.5%, *p* = 0.02). We have also observed a tendency to more time spent in hypoglycemia level 1 (8.0 vs. 5.0%, *p* = 0.05) but not in level 2 hypoglycemia (2.0 vs. 1.0%, *p* = 0.78) in the LCD group. Time in hyperglycemia was significantly lower in the LCD group with 17.0 vs. 20.0% (*p* = 0.04) at level 1 hyperglycemia and 2.0 vs. 3.0% (*p* = 0.04) at level 2 hyperglycemia, respectively. The average glycemia was also significantly lower in the LCD group (7.0 vs. 7.9 mmol/L, *p* = 0.02), as was the standard deviation of glycemia excursions (2.6 vs. 3.2 mmol/L, *p* = 0.03). The coefficient of variation was not different between the groups (37.4 vs. 39.5%, *p* = 0.60).

The groups did not differ in their body height SDS (−0.4 vs. 0.3, *p* = 0.35), body weight SDS (0.5 vs. 0.4, *p* = 0.68) nor BMI SDS (0.5 vs. 0.5, *p* = 0.62). The LCD group showed lower percentile of systolic blood pressure (43 vs. 73, *p* = 0.03) but the diastolic pressure percentile was similar in both groups (61 vs. 76, *p* = 0.17). The values of the lipid spectrum were not statistically different between the groups.

### 3.3. Questions Specific for the LCD Group

The results are shown in Table 2. The subjects’ motivation for LCD initiation was mostly the desire for better diabetes control and healthy lifestyle with weight reduction being also given as one of the more frequent reasons. In the vast majority of the subjects, the parents/caregivers or the subjects themselves decided to start with the LCD. More than a third of the subjects did not consult their diabetologist prior to this decision. Subjects generally sought the Internet, books and other parents of children with T1D for information on the LCD, only a minority asked their diabetologist. The subjects referred better T1D control, lower insulin dose and weight reduction as the positive changes brought about by the LCD. On the other hand, they reported more time and money spent for meal preparation, more frequent hypoglycemia and fatigue. The decision to terminate the LCD came equally often from the parents/caregivers as the subjects themselves. The main reasons for LCD termination were the non-compliance of the child and the observed side-effects (fatigue, frequent hypoglycemia).

### 3.4. Comparison of the VLCD Sub-Group to LCD Subjects

The subjects who followed the very low-carbohydrate diet (*N* = 5) did not significantly differ from the LCD subjects in age, duration of T1D, sex nor type of the treatment. They kept the diet longer than the subject on LCD (3.0 vs. 0.9 years, *p* = 0.004). They had significantly lower daily carbohydrate intake (35 vs. 100 g, *p* < 0.001) but did not differ in insulin doses nor their disease control as assessed by HbA_1_c or CGM values. The subjects on the VLCD had tendency toward higher body weight SDS (1.8 vs. 0.4, *p* = 0.22) and BMI SDS (1.4 vs. 0.4, *p* = 0.21) but the differences did not reach statistical significance. The VLCD subjects also showed disturbed lipid spectrum with marginally higher total cholesterol (5.3 vs. 4.7 mmol/L, *p* = 0.05) and lower HDL cholesterol (1.3 vs. 1.5 mmol/L, *p* = 0.08). All analyses can be found in Appendix A.

## 4. Conclusions

The results of the study on a representative cohort indicate that the use of the LCD is quite commonplace among children/adolescents with T1D with 38.7% having the experience with carbohydrate reduction and 5.8% currently keeping the LCD. Their main motivation for the initiation of the diet was to improve their glycemic curves and we have shown that children with T1D on the LCD tend to have excellent disease control at the cost of slightly higher time spent in hypoglycemia. The subjects or their parents/caregivers are mostly seeking non-professional sources for advice on this nutritional intervention, a fact that should not be ignored since restrictive diets tend to imbalanced nutritional intake with possibly harmful consequences [17].

The comparison between the LCD subjects and a well-matched control group revealed that the LCD group has excellent disease control with medians for TIR and time in hyperglycemia falling well into the recommended zones. Lower standard deviation of glycemia excursions also suggests more stable glycemia in the LCD group, possibly due to lower postprandial excursions, which were mentioned as one of the possible effects earlier [14,31]. The findings of lower blood pressure in LCD subjects were reported on an adult cohort in study by Ahola [31]. Our findings of lower systolic pressure among our LCD subjects might be linked to lower insulin doses. Higher insulinemia was found to be connected to higher increase in blood pressure in children adolescents over the course of 6 years [32] and hyperinsulinemia was also found to be an independent risk factor for the development of hypertension [33].

Among the drawbacks might be the already described tendency to increased time in hypoglycemia [34], yet the observed differences were bordering on significance only in level 1 hypoglycemia and no indication suggests increased frequency of the more severe level 2. On the other hand, hypoglycemia was one of the more common reported reasons for LCD discontinuation and two of the subjects who terminated the LCD had an episode of severe hypoglycemia which brings the findings of Ranjan [20], who described blunted response of glucagon to hypoglycemia on LCD to the front. We hypothesize that more frequent and severe hypoglycemia can occur shortly after LCD initiation as a result of inadequate lowering of insulin doses as the ones who mentioned hypoglycemia as the reason for LCD termination were the ones who ended with LCD early. No disturbances were noted in the lipid profile of the subjects which is in contrast to case-series by de Bock et al. who describes disturbed lipid spectrum in four of the six cases [17]. The probable explanation for this difference is that the patients described had daily carbohydrate intake below 50 g, whereas the median in our cohort was 96.5 g. In support of this, our analysis of the very low-carbohydrate subset within our cohort also identified a tendency towards higher total cholesterol and lower HDL as compared to the regular LCD.

Vast majority of the subjects who follow the LCD has the follow-up center in the capital Prague rather than in the regional city Brno. This might be linked to higher per capita income in Prague [35] as well as the higher knowledge on special diets and nutrition in larger urban areas [36]. More than two thirds of subjects currently on LCD were female, possibly due to the higher observed tendency for dietary interventions in females with T1D [37]. Another possible explanation is that the LCD group includes teen-aged girls whose main reason for LCD is to decrease insulin dose and consequently reduce body weight.

Despite generally not being discouraged from LCD by the diabetologists when consulted, the patients often did not discuss their decision to start LCD with them. Instead, they relied on unofficial sources like Facebook groups or pages that promote LCD in adults without T1D. This might possibly have dire consequences as the effects of carbohydrate restriction on children let alone with T1D are not fully explored. We would therefore promote a cautionary position with emphasis on the possible risks as well as the benefits of the LCD. Our study did not focus on the added psycho-social burden of the restrictive diets [38], yet some subjects reported increased family and school related conflict, creating a possibility for future research.

Among the strengths of this study are its considerably high response rate and relatively wide study population of children/adolescents with T1D. The data for the ČENDA Registry are collected quarterly and thus provide very accurate and recent anthropometric and metabolic information on the patients. Furthermore, the compared groups were tightly matched to minimize potential bias.

Among weaknesses of this study is the fact that it omitted the smaller centers for diabetes care in the Czech Republic, where the prevalence of LCD would presumably be lower. Due to lack of contact information, we did reach only 83% of the children followed in the study centers. Furthermore, the carbohydrate intake in our study was self-reported by the subjects or their parents and lacked information on daily protein and fat intake. In addition, the design does not allow to infer any causality—for that, an intervention study is underway. Anthropometric features like body weight and height as well as BMI would also need to be followed longitudinally to assess.

Our study underlines the fact that carbohydrate reduction is considerably popular in children/adolescents with T1D. Patients often seek non-professional sources of information and do not consult their diabetologists with their decisions. The observed positive effects cannot be overestimated and until proper prospective trials are conducted, we should inform our patients of potential risks, especially the increased risk of hypoglycemia and possible disturbance in the lipid spectrum in the case of VLCD. We should therefore individualize the treatment and seek safer ways to optimize their glycemia together.

## Figures and Tables

**Figure 1 nutrients-13-03903-f001:**
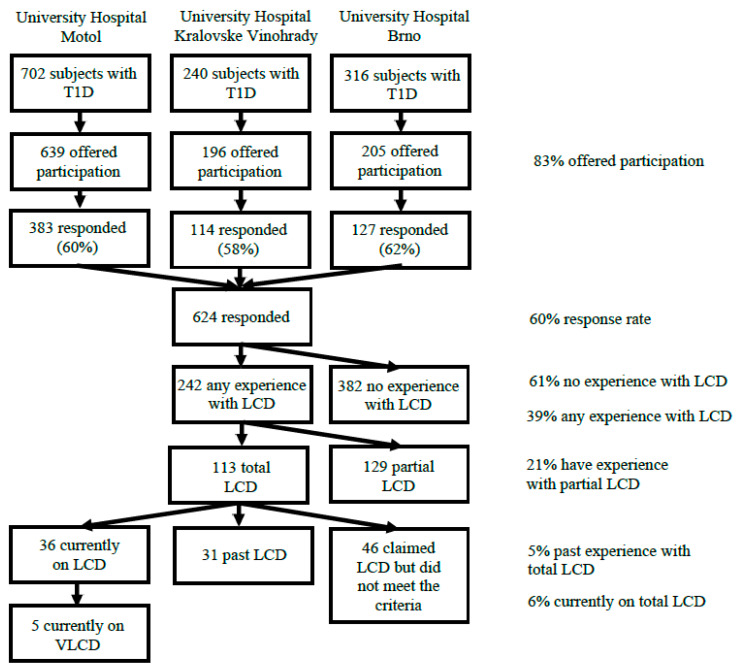
The study flowchart. LCD—low-carbohydrate diet, T1D—type 1 diabetes, VLCD—very low-carbohydrate diet.

**Table 1 nutrients-13-03903-t001:** Descriptive characteristics of the currently on total low-carbohydrate diet group, data are shown as median (IQR).

	Low-Carbohydrate Diet	Non-LCD Diet	*p*-Value
Subjects	*N* = 36	*N* = 36	
Demographics			
Age at survey collection (years)	11.9 (6.1)	11.8 (5.6)	0.03 * (0.06)
Sex	F = 25 (69.4%) M = 11 (30.6%)	F = 25 (69.4%) M = 11 (30.6%)	-
Age at T1D onset (years)	8.0 (7.1)	8.4 (5.6)	0.11 (0.21)
T1D duration (years)	3.2 (2.7)	3.7 (3.5)	0.74 (0.87)
Anthropometric data			
Body height (cm)			0.54 (0.57)
Body height SDS	−0.4 (1.4)	0.3 (1.2)	0.35 (0.42)
Body weight (kg)			0.62 (0.66)
Body weight SDS	0.5 (1.2)	0.4 (1.0)	0.68 (0.72)
Body mass index SDS	0.5 (1.3)	0.5 (0.9)	0.62 (0.68)
Systolic arterial blood pressure (centile)	43 (63.0)	73.5 (40.8)	0.03 * (0.008 **)
Diastolic arterial blood pressure (centile)	61 (32.5)	76.5 (39.8)	0.17 (0.08)
LCD data			
Daily carbohydrate intake (g)	96.5 (42)	170 (39.8)	<0.001 *** (<0.001 ***)
Age at LCD start (years)	11.2	-	-
LCD duration (years)	1.1	-	-
LCD type	LCD = 31 (86.1%) VLCD = 5 (13.8%)	-	-
Treatment and T1D control			
Treatment type	MDI = 27 (75%) CSII = 9 (25%)	MDI = 27 (75%) CSII = 9 (25%)	-
Bolus insulin (units daily)	10.0 (10.4)	21.5 (16.8)	<0.001 *** (<0.001 ***)
Basal insulin (units daily)	12.0 (12.7)	13.5 (11.6)	0.76 (0.60)
Total insulin daily dose (units/kg/day)	0.6 (0.3)	0.8 (0.3)	<0.001 *** (<0.001 ***)
Last HbA1c (mmol/mol)	45.0 (9.5)	49.5 (15.2)	0.11 (0.26)
Last HbA1c (%)	6.3 (3.0)	6.7 (3.6)	0.11 (0.26)
Average HbA1c during the last year (mmol/mol)	47.9 (10.3)	50.9 (10.6)	0.05 (0.11)
Average HbA1c during the last year (%)	6.5 (3.1)	6.8 (3.1)	0.05 (0.11)
CGM data			
CGM use	Yes = 35 (97.2%) No = 1 (2.8%)	Yes = 34 (94.4%) No = 2 (5.6%)	0.75 (0.75)
Time in range 3.9–10.0 mmol/L (%)	74.0 (14.5)	66.5 (16.4)	0.02 * (0.05)
Time below 3.9 mmol/L (%)	8.0 (8.0)	5.0 (6.0)	0.05 (0.25)
Time below 3.0 mmol/L (%)	2.0 (3.0)	1.0 (4.5)	0.78 (0.89)
Time above 10.0 mmol/L (%)	17.0 (15.9)	20.0 (9.0)	0.04 * (0.20)
Time above 13.9 mmol/L (%)	2.0 (4.0)	3.0 (5.0)	0.04 * (0.07)
Average glycemia (mmol/L)	7.0 (1.2)	7.9 (2.1)	0.02 * (0.05)
Standard deviation of glycemia	2.6 (1.0)	3.2 (0.8)	0.03 * (0.07)
Coefficient of variation (%)	37.4 (8.0)	39.5 (11.8)	0.60 (0.80)
Lipid spectrum			
Total cholesterol (mmol/L)	4.8 (0.8)	4.7 (1.4)	0.55 (0.83)
Triglycerides (mmol/L)	0.9 (0.5)	0.9 (0.6)	0.29 (0.08)
HDL cholesterol (mmol/L)	1.5 (0.5)	1.7 (0.4)	0.23 (0.46)
LDL cholesterol (mmol/L)	2.6 (0.9)	2.7 (1.1)	0.57 (0.98)

CGM = continuous glucose monitoring, CSII = continuous subcutaneous insulin infusion, F = female, IQR = interquartile ratio, HDL = high density lipoprotein, LCD = low-carbohydrate diet, LDL = low density lipoprotein, M = male, MDI = multiple daily injection, SDS = standard deviation score, T1D = type 1 diabetes, VLCD = very low-carbohydrate diet. The *p*-values in brackets show the significance with the VLCD subgroup (*N* = 5) excluded from the analysis (*N* = 31), * *p* < 0.05, ** *p* < 0.01, *** *p* < 0.001.

**Table 2 nutrients-13-03903-t002:** Survey responses of the subjects with present or past experience with LCD (*N* = 67).

Question	Answer	*N*	%
Who initiated carbohydrate restriction?	The child/adolescent with T1D	18/67	26.8
	The parent/caregiver	42/67	62.7
	The diabetologist	6/67	8.9
	No answer	1/67	1.5
Did you consult your intention to reduce carbohydrate intake with your diabetologist?	No	26/67	38.8
	Yes, he/she supported us	18/67	26.9
	Yes, he did not support us nor discouraged us but explained the risks and benefits	17/67	25.4
	Yes, he/she discouraged us	4/67	6.0
What was the reason for carbohydrate restriction? *	Better T1D control	46/67	68.7
	Lower insulin dose	20/67	29.9
	Reduction of body weight	12/67	17.9
	Healthy lifestyle	40/67	59.7
What were your sources of information on carbohydrate restriction? *	Internet	42/67	62.7
	Books	23/67	34.3
	Other families with T1D children	27/67	40.3
	Diabetologist	16/67	23.9
Have you noticed any changes after the initiation of carbohydrate restriction? If so, which? *	No	9/67	13.4
	Better T1D control	32/67	47.8
	Lower insulin dose	40/67	59.7
	Body weight reduction	21/67	31.3
	Increased fatigue	7/67	10.4
	More frequent hypoglycemia	12/67	17.9
	School-related conflict	4/67	6.0
	Conflicts with diabetologist	1/67	1.5
	Family conflict	7/67	10.4
	Increased costs for meal preparation	13/67	19.4
	Increased time for meal preparation	15/67	22.4
	Hunger	1/67	1.5
Did you/your child find low-carbohydrate meals tasty?	Yes, or mostly yes	29/67	43.3
	Depending on the meal	26/67	38.8
	No, or mostly no	11/67	16.4
Did you/your child with T1D have severe hypoglycemia requiring hospitalization during the time you reduced carbohydrates?	Yes	2/67	3.0
	No	65/67	97.0
Did you/your child with T1D have diabetic ketoacidosis requiring hospitalization during the time you reduced carbohydrates?	Yes	0/67	0.0
	No	67/67	100.0
Would you recommend carbohydrate restriction to other children/adolescents with T1D?	Yes	49/67	73.1
	No	18/67	26.9
Who initiated the termination of carbohydrate restriction? ^†^	The child/adolescents with T1D	14/31	45.2
	The parent/caregiver	14/31	45.2
	The diabetologist	3/31	9.7
What were the reasons for the termination of carbohydrate restriction? *^,†^	It did not fulfill our expectations	13/31	41.9
	Non-adherence on the side of child/adolescent with T1D	16/31	51.6
	Side-effects (fatigue, frequent hypoglycemia)	7/31	22.6
	Financial costs	2/31	6.5
	Time consumption	4/31	12.9
	School-related conflict	2/31	6.5
	Family conflict	1/31	3.2
	No answer	2/31	6.5

* ubjects were allowed to answer multiple times to this question. ^†^ Only subjects who have terminated LCD answered this question (*N* = 31).

## Data Availability

All data used for the analysis in this article are available on request from the authors.

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
