# Peer review of "Low-Carbohydrate Diet among Children with Type 1 Diabetes: A Multi-Center Study"

_nutrients, 2021, doi:10.3390/nu13113903_

Round 1
Reviewer 1 Report
Interesting study assesing the frequency of the low-carbohydrate diet (LCD) use in a real life of a great group of pediatric patients with type 1 diabetes (T1D).
Major objections concerning the study and prepared paper:
- Definition of LC diet (LCD) applied in the study group of children is not proper. LCD given by authors and defined as daily carbohydrate intake < 130 g or < 26% of total energy intake ( 8) is typical for adults. The same concerns very low carbohydrate diet (VLCD) defined by authors. Carbohydretes intake < 50 g per day or <10% energy intake is the definition of VLCD given for adults.
- LCD/VLC diet in chidren is different and should be adjusted to the age of them ( 8), and may be defined by percentage of daily energy intake or by carbohydrate intake (lower than for adults), or in g/kg body weight/day. It is of great importance, as the mean age of investigated group was approx. 12 years.
- Proper difinition may change qualification of the patients an may have significant impact on the results of the study.
- Reassessement of the results should be done after making changes in the definition of LCD and VLC diets.
- Definition of LCD given in the Introduction section (line 53-54) and in the Research Design and Methods section (line 108-109) can not be found in cited ref. No 9. Proper citation has to be done.
- The weight of investigated chidren is not stated (Table 1), so it is difficult to assess daily carbohydrate intake adjusted for body weight.
- LCD diet may be used as either normal-fat/high-protein diet or high-fat/normal-protein Metabolic impact of these two different diets (especially on lipids) may be quite different. The survey did not search for daily protein/fat content.
Minor objections:
- Table 1: Some abbrevations are not explained (SSD), Age at LCD start and LCD duration – (year/month ?) are not written.
- Some references are not cited properly (e.g ref. 8 – false year, ref. 9 – lack of authors, lacking name of journal).
- Figure 1 with a flowchart (Results section, page 7) should be inserted much earlier in the Methods section.
- Figure 1. Response rate 60% is ok, but this number is discordant with total percentage given fo 3 different hospitals (55%-48%-40%). These hospitals response range should be stated for patients/families offered participation (n=1040).
- Table 2 should be presented in a proper order (after Table 1.) in a proper section (Results).
- ESM (?) Table 1 is cited but not inserted (?).
Reviewer 2 Report
This report combines survey data with clinical data collected from medical records. The report offers interesting clinical comparisons between self-reported LCD adherence and non-adherence in children with T1D. However, some items need attention for transparency and proper data interpretation.
The authors report that the survey respondents had previously signed a written consent to allow anonymized secondary research using their clinical data. It is not clear if the survey respondents gave permission for their survey responses to be matched to their clinical data. The authors will need to clarify that this matching (which required identified data) was permitted by the respondents.
The study offers interesting comparisons between those currently following a LCD and closely matched patients who never followed a LCD. However, 14% of those in the former group were following a VLCD which may have influenced the LCD data reported in Table 1 (either favorably or not favorably). The authors state ‘increased risk of hypoglycemia and possible disturbance in the lipid spectrum’ in VLCD – it is surprising that they collapsed these two groups for the analyses. Are the data different with these 5 patients removed from the analyses?
The Table 1 heading ‘standard diet’ is not justified. ‘Non-LCD’ diet would be a better descriptor.
The conclusion that ‘our study underlines the fact that the LCD is considerably popular’ seems to contradict the data that show only 5.8% of respondents were currently on a total LCD diet and 5% reported previously following a total LCD diet. Can the authors justify this conclusion?
Round 2
Reviewer 2 Report
Thank you for making the suggested changes. For Table 1 please retain the original table and add a footnote regarding the changes in significance when the VLCD group is excluded.
Author Response
Thank you for the comment. The table has been updated and the footnote added.